# Residual Heat Effect on the Melt Pool Geometry during the Laser Powder Bed Fusion Process

**Subin Shrestha * and Kevin Chou**

J.B. Speed School of Engineering, University of Louisville, Louisville, KY 40292, USA
* Correspondence: subinsht@gmail.com

**Abstract:** The continuous back-and-forth melting of the powder bed during the laser powder bed fusion (LPBF) process leads to the development of residual heat, which affects the melt pool geometry as the laser scan progresses. The magnitude of the residual heat depends on the scan length, hatch spacing, location on the track, etc. In this regard, back-and-forth raster scanning was performed to investigate the effect of the scan length and hatch spacing on the melt pool size at different locations along the laser travel direction. Multi-track specimens with different scan lengths (0.5 mm, 1 mm, and 1.5 mm) were fabricated using 195 W laser power, three scan speeds (375 mm/s, 750 mm/s, and 1500 mm/s), and two hatch spacings (80 μm and 120 μm). A white light interferometer was used to analyze the surface morphologies of the fabricated samples, and metallography was performed to observe the melt pool boundary. The melt pool boundary obtained at different locations revealed that the effect of the residual heat was maximal in the laser-turn region. In addition, a powder scale numerical model was developed to investigate the effect of temperature distribution on the melt pool geometry. The numerical results show that the laser-turn region was most affected by the residual heat, as the melt pool from the two tracks merged. The depth of the melt pool increased with increasing track numbers, while the track height decreased. The addition of a second layer of powder showed that the inherent surface variation in the first layer leads to the difference in the actual layer thickness of the second layer.

**Keywords:** additive manufacturing; laser powder bed fusion; porosity; numerical modelling; melt pool

## 1. Introduction

Laser powder bed fusion (LPBF) is an additive manufacturing (AM) process that uses a laser to melt the metallic powder particles layer by layer. Layer-wise manufacturing enables the fabrication of very complex geometries, which include lattice structures [1], topology-optimized structures [2], etc. Such parts often consist of thin features formed by using a few scan tracks, and these thin features are formed with very short scan lengths.

The laser scanning is performed with vectors that are parallel to each other in the LPBF process to fabricate the parts [3]. The smaller areas have shorter scan lengths, which result in high temperatures due to a short cool-down time. Yadroitsev and Smurov [4] studied the effect of hatch spacing on surface morphology. Their analysis of the cross sections of the tracks fabricated on the build substrate showed the difference in the track height during the multi-track formation. The track at the beginning of the sequence was higher than the subsequent tracks. Pupo et al. [5] concluded that the surface morphology in a multi-track experiment is strongly affected by the hatch spacing due to heat accumulation in the melted zone. Moreover, heat accumulation during the multi-track scan affects the subsequent size of the melt pool [6]. Hu et al. [7] investigated the evolution of the contact angle during multi-track scanning. The cross-sectional model showed that the contact angle decreased initially and then became stable with the increasing track number. The authors concluded that the trend is not affected by the changes in scan speed or hatch spacing, rather it is

determined by the fluid flow driving forces and the lifetime of the melt pool. In this regard, numerical models were used to predict the fluid flow and melt pool size.

Lee and Zhang [8] utilized the powder scale numerical model to understand the effect of the first scan track on the melt pool of the second track. The melt pool shape for the first track was symmetric, while an asymmetric molten pool was obtained for the second track. This is due to the residual heat contained in the first track while depositing the second track. The preheating from the first track leads to the expansion of the second track towards the first track. A more symmetric melt pool can be obtained by increasing the hatch spacing, which results in the successive melt pool forming away from the heat-affected zone due to the melting of the previous track [9]. Criales et al. [10] performed a multi-track simulation for the LPBF process and observed an increase in the melt pool size in the subsequent tracks due to the heat-affected zones from the previous tracks. Bayat et al. [11] developed a numerical model to investigate the formation of the lack-of-fusion voids in Inconel 625 during multi-track/multi-layer LPBF. The study suggested that the lack-of-fusion pores, which were mostly elongated in the direction parallel to the scanning tracks, were mainly found in the lower layers due to the lower thermal energy compared to the higher levels. Gu et al. [12] studied the melt pool formation in multi-track, multi-layer, and multi-material during the LPBF process.

The difference in the surface height between the successive tracks is apparent in the LPBF process, and such variations in the surface morphology may lead to the internal porous structure during layer-wise fabrication [4]. Additionally, the tracks formed have transient regions that will affect the quality of the part, especially with the shorter scan vectors [13]. On the other hand, increasing the scan length may increase the residual stress, which is also undesirable [14]. The residual heat effect is critical for thin features and there has been no research on the relationship between the residual heat and scan lengths. Hence, this research provides a comprehensive study of the residual heat effect on the surface morphology and melt pool through experiment and simulation. Furthermore, the presence of transient regions, which is often neglected, is significant for shorter scan lengths or thin feature sizes. In this study, EOS M270 was used to fabricate multi-track and multi-layer samples using Inconel 625. Three back-and-forth scan tracks using different levels of scan speed, hatch spacing, and scan lengths were used to fabricate single and multi-layer samples. The surface profiles of the multi-track and multi-layer samples were obtained using the white light interferometer. The variation in the surface height due to the back-and-forth scanning was analyzed. Further, the transverse metallography was performed in the laser turn-around regions and the center of the tracks to observe the variation in the melt pool depth. In addition, a powder scale numerical model was developed to understand the effect of residual heat on the melt pool at scan lengths of 0.5 mm and 1 mm.

## 2. Experimental Approach

Table 1 shows the experimental design used to form the multi-track and multi-layer samples. In this study, a powder size distribution of 15–45 μm was used. Three levels of scan speed, two levels of hatch spacing, and three levels of scan lengths were used. Hence, multi-track and multi-layer samples with a combination of 18 parameters were fabricated for each. Three tracks were formed with back-and-forth scanning to investigate the effect of residual heat on the melt pool, while two layers were formed for the multi-layer study. Three scan speeds were utilized to include three modes of melting: incomplete melting, conduction melting, and keyhole melting. Two hatch spacings, which is the distance between two successive laser paths, were used to observe the effect of residual heat on the track overlap and the surface morphology. The effect of the first-layer surface quality on the second-layer formation was also of interest.

Figure 1 illustrates the scanning pattern utilized in this study. Martin et al. [15] investigated the pore formation mechanism at the laser-turn point during the LPBF process using in situ X-ray imaging and multi-physics simulation. The change in the laser scan velocity at the turn points leads to the formation of a deeper keyhole, and the pores are

formed due to the collapse of the keyhole depressions [15]. In this study, the skywriting setting was used, which ensures that the acceleration and deceleration of the laser occur outside of the hatch lines, thus maintaining the constant energy input inside the scan lines. The laser jumps the distance of the hatch spacing to form successive tracks, and the same pattern is used for the second layer, that is, the first track in the second layer is formed on top of track 1 of the first layer.

**Table 1.** Experimental design used to fabricate multi-track samples.

| Factor | Value |
|---|---|
| Laser Power (W) | 195 |
| Laser spot size (μm) | 100 |
| Scan speed (mm/s) | 375, 750, 1500 |
| Hatch spacing (μm) | 80, 120 |
| Scan length (mm) | 0.5, 1, 1.5 |

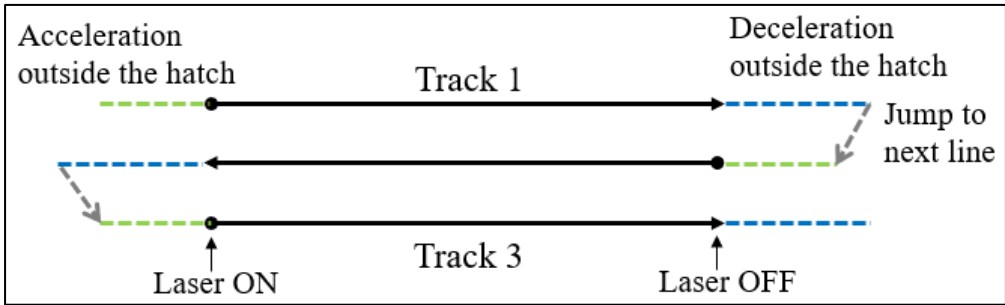

**Figure 1.** Schematic of the constant energy input in the scan lines during raster scanning.

Figure 2a shows the sample design. EOS M270 was used to fabricate multi-track and multi-layer samples with a layer thickness of 40 μm, and the fabricated samples are shown in Figure 2b. The semi-cylinder base was fabricated with 195 W laser power, 750 mm/s hatch spacing, and 120 μm hatch spacing, while the multi-track samples were fabricated on top of the base using the parameters listed in Table 1. The semi-cylinder base was used to reduce the contact area with the support. The block support was used to support the sample, and the samples were detached from the build platform after the fabrication, and the support was removed by polishing. The multi-track and multi-layer samples were fabricated at a height of 4.6 mm, that is, after 115 layers.

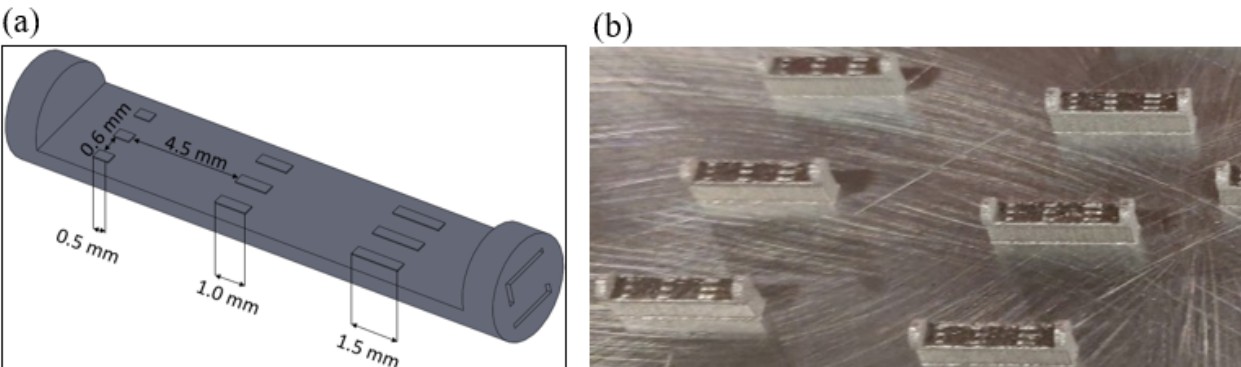

**Figure 2.** (**a**) Single layer designed on top of the semi-cylindrical base, and (**b**) single layer samples fabricated using EOS M270.

A white light interferometer was used to obtain the surface morphologies of the multi-track and multi-layer samples. In addition, metallography was performed to determine the mode of melting and observe the variations in the transverse melt pool boundaries at different regions along the scan direction.

## 3. Numerical Approach

### 3.1. Discrete Element Method

During the LPBF process, the build platform is lowered by a layer thickness, and a layer of powder is spread using a roller or a recoater blade depending on the LPBF system. EOS M270 utilizes a recoater blade to spread the powder, and Figure 3 shows the schematic of the major components involved in the layering process. It is important to model the process of powder spreading to obtain a representative powder bed distribution. Hence, an open-source DEM code called LIGGGHTS was used to simulate the powder-spreading process [16]. The powder size distribution of 15–45 µm was used in this study. The powder particles generated in the dispenser were spread over the build platform by the recoater blade, as shown in Figure 4. The gap between the recoater and the build platform determines the powder distribution. However, the actual thickness of the layer of the powder may vary according to the build height due to the inherent shrinkage during the building process. Several studies have mentioned that the theoretical steady-state powder layer height is equivalent to the layer thickness divided by the packing density [17], and the layer height reaches a steady state after 10 layers or so. Therefore, assuming that the powder packing density is 50%, the steady state layer thickness is 80 µm. As the single tracks were formed after reaching a steady state in the experiment, a gap distance of 80 µm was used between the build platform and the recoater blade for the first layer in the DEM simulation.

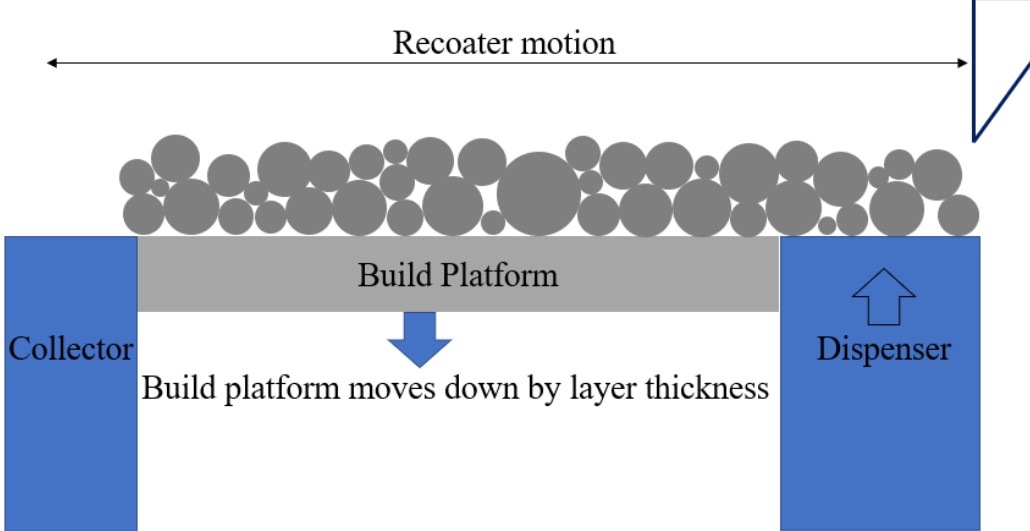

**Figure 3.** The powder-spreading process during LPBF process.

For the DEM simulation of the second layer, the surface formed after scanning the first layer was utilized. The powder bed from the first layer was lowered by 80 µm, which is the actual thickness of the first layer. However, the gap between the recoater blade and the base was reduced to 40 µm for the second layer as the shrinkage was accounted for during the first layer scanning simulation. This method enabled the investigation of the variation in the powder distribution and layer thickness across the second layer due to the inherent surface roughness in the first layer. Figure 4 depicts the variation in the gap between the first layer and second layer after the powder-spreading process.

### 3.2. Thermo-Fluid Simulation

The layer of powder particles on top of the build platform was obtained from the DEM simulation, which was then imported to the FLOW-3D commercial software to perform the thermo-fluid simulation. Figure 5 shows the simulation domain used for a 0.5 mm scan length. Two scan lengths were simulated, 0.5 mm and 1 mm, and the length of the simulation domain was modified based on the scan length. Further, the simulation domain was divided into two zones: the inner zone has fine hexahedral mesh (5 μm) where laser scanning is performed, and the outer zone has coarse mesh where the thermal gradient is low.

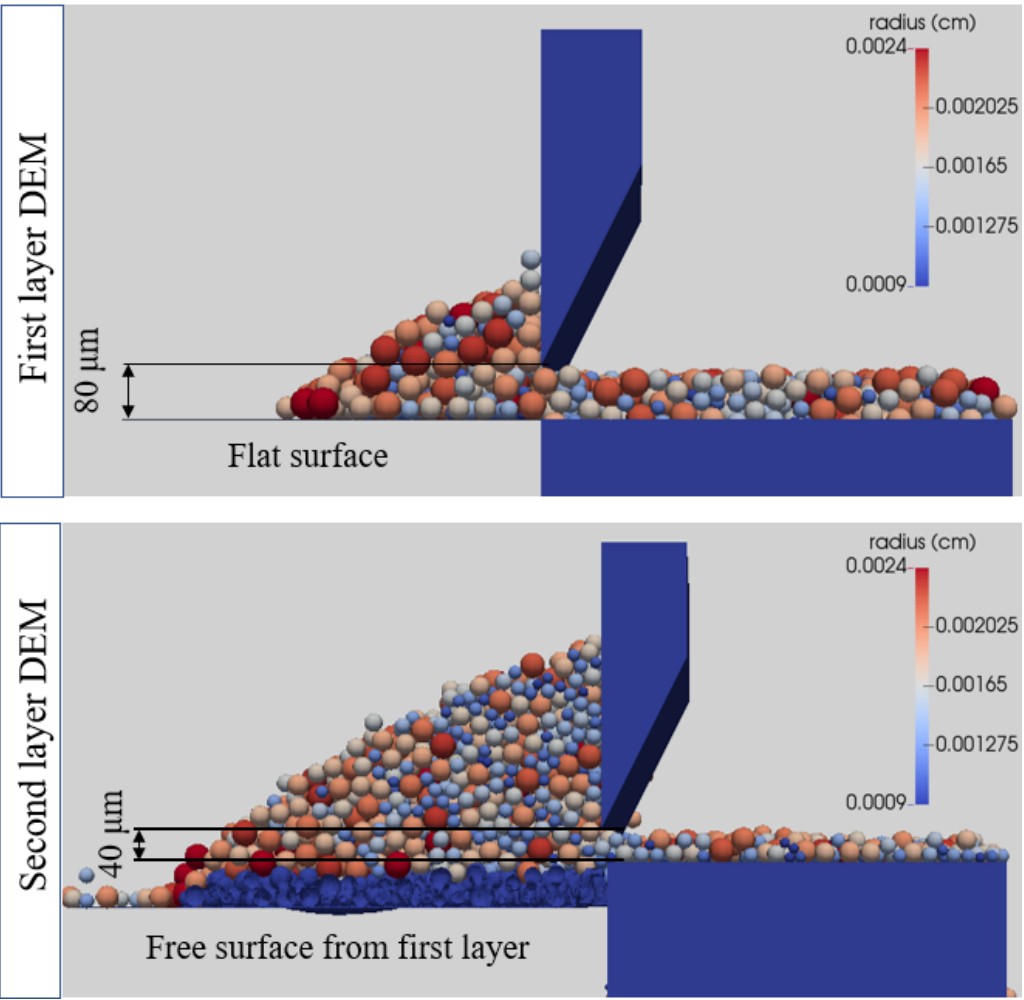

**Figure 4.** Consideration of actual powder height for first layer and second layer DEM simulation.

The temperature-dependent IN625 material properties were assigned to the powder layer and solid substrate. Figure 6 shows the temperature-dependent conductivity, specific heat capacity, and density of IN625. Moreover, the other material properties used in the simulation are listed in Table 2 [18]. A Gaussian laser was used to perform the back-and-forth scanning and investigate the effect of scan length on the melt pool. After the completion of the first layer, the free surface was used to simulate the second layer of powder spreading in LIGGGHTS. The obtained second layer powder was again imported to the thermo-fluid model, and finally, back-and-forth scanning was performed.

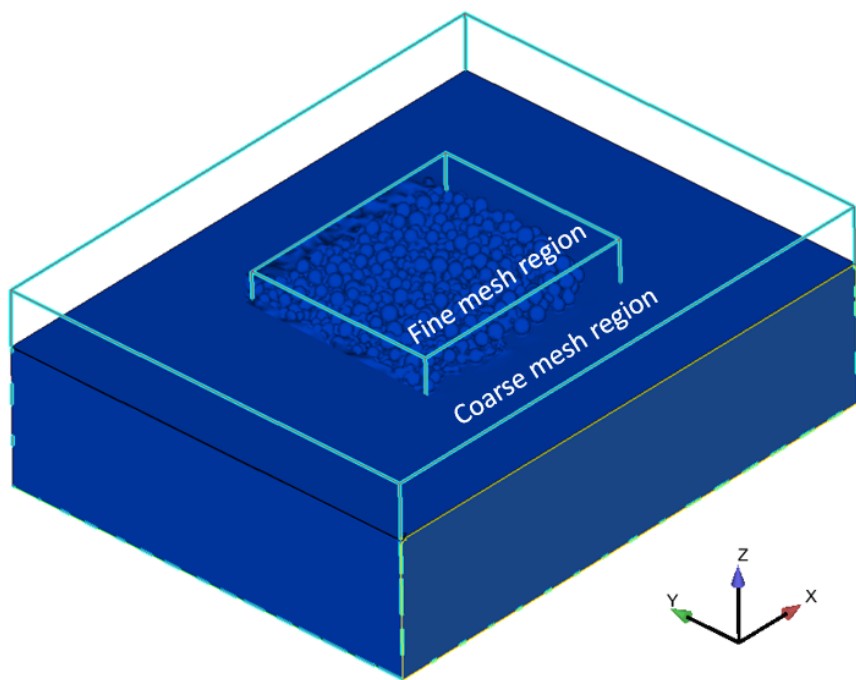

**Figure 5.** Simulation domain used for 0.5 mm scan length simulation.

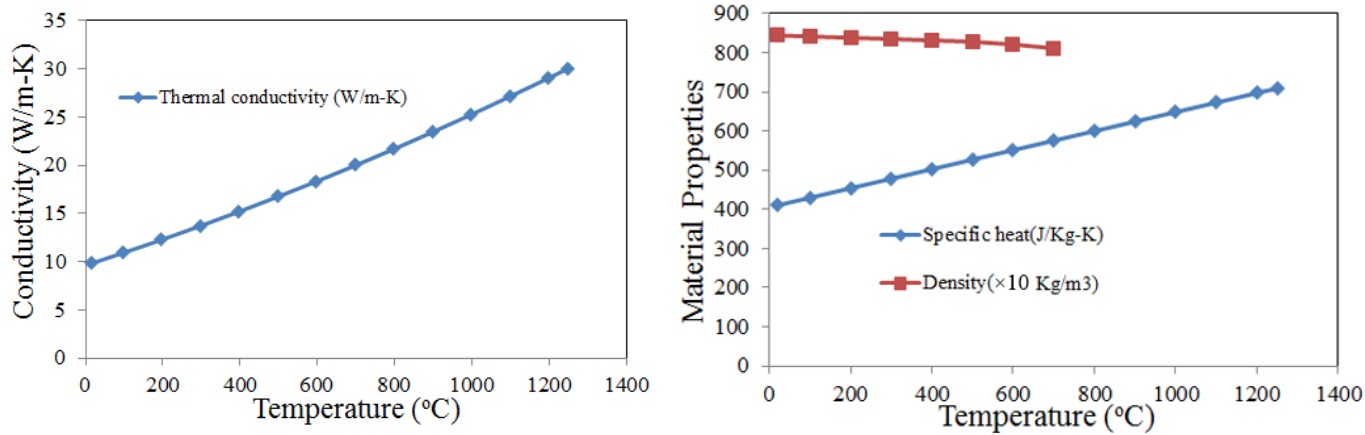

**Figure 6.** Temperature-dependent material properties of Inconel 625 [19].

**Table 2.** Properties of IN625 in a thermo-fluid simulation.

| Parameters | Values |
|---|---|
| Solidus temperature, $T_S$ | 1563 K |
| Liquidus temperature, $T_L$ | 1623 K |
| Boiling temperature, $T_v$ | 3188 K |
| Viscosity | 0.007 kg/m/s |
| Surface tension | 1.8 N/m |
| Surface tension gradient | $-2 \times 10^{-5}$ N/m/K |

## 4. Results and Discussion

### 4.1. Experimental Results

4.1.1. Surface Measurements

Figure 7 presents the surface morphologies of multi-track samples fabricated with 195 W laser power, 750 mm/s scan speed, and 120 μm hatch spacing at different scan lengths. The difference in the magnitude of the heat accumulation clearly affected the

surface morphology. The effect of the residual heat was higher for the 0.5 mm scan length as the three tracks resulted in a single bump and the individual tracks could not be identified. As the length increased, the tracks became distinct, and the transient and steady region was more obvious for 1.5 mm. Although the difference in the profile was evident, the maximum height difference between the track surface and the base was comparable; this was around 120 μm for all scan lengths. Additionally, the maximum track height was seen at the beginning of the track due to the formation of a hump. The hump was formed due to the backward surge of the melt pool at the laser turn-on region [20]. It was also observed that the minimum surface height mostly lies at the end of the track. This is due to the formation of a depression in the laser turn-off region. In the laser-turn regions, the melt pool formed during the scanning of the previous track may not fully solidify while another track is being scanned, resulting in the merging of the two tracks. Hence, it was also observed that the laser-turning regions have continuity as if the laser itself turned to form an arc although the laser jumped between the tracks.

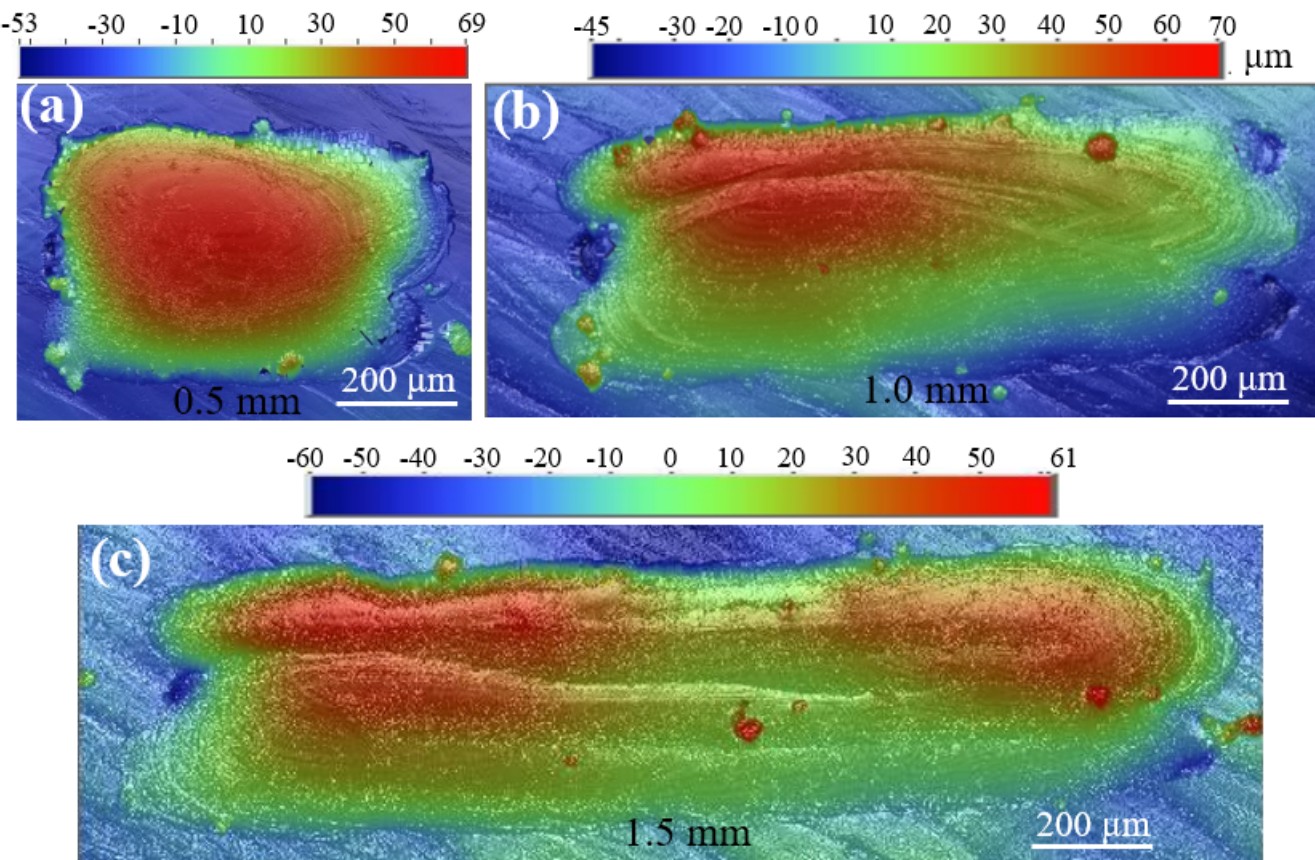

**Figure 7.** Multi−track profiles formed with 195 W, 750 mm/s, 120 μm hatch spacing and (**a**) 0.5 mm, (**b**) 1.0 mm, and (**c**) 2.0 mm scan length.

The surface formed with the same parameters was compared between multi-track and multi-layer samples, as presented in Figure 8. The transverse profile was obtained to include the region with the maximum surface height as indicated in Figure 8. The surface height gradually decreased with the increasing number of tracks. The 2D profile shows that the maximum surface height for the single layer is around 75 μm, while the maximum height of the two-layer sample is 150 μm. The maximum surface height was measured from the surface over which the tracks were deposited. The single-track experiments showed that the surrounding particles were depleted as the powder particles were entrained in a shear flow of gas driven by a metal vapor jet at the melt track [21]. This denudation phenomenon increases the surface height of the first track significantly. On the other hand,

the depletion of the powder particles in the surrounding results in lower powder mass for the next track. Hence, the track height decreased for the second track.

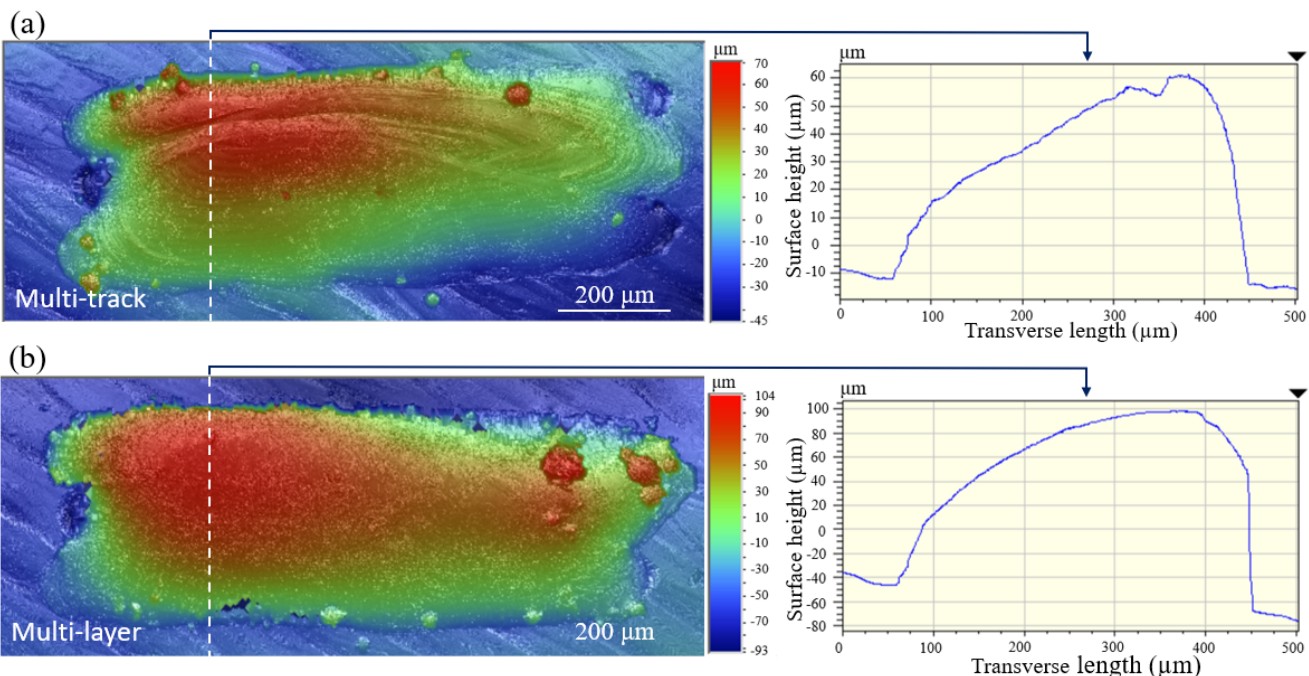

**Figure 8.** Surface morphology obtained from (**a**) multi−track, single layer, and (**b**) multi−track, multi-layer samples (195 W, 750 mm/s and 120 µm).

4.1.2. Metallography

The metallography was performed on 0.5 mm and 1 mm long multi-track samples formed with the parameters, 195 W and 120 µm, at different scan speeds. The melt pool boundaries at the start, middle, and end of the tracks were obtained and are shown in Figure 9a. Figure 9b–d show the melt pool boundary obtained in three regions. The sequence of the track was the same for all the images, as shown in Figure 9b for 1500 mm/s. The left-most track is the first laser path, and the one to the right is the third track. There is a significant overlap between the tracks for conduction mode (750 mm/s) and keyhole mode (375 mm/s) melting. However, there was no effect on the higher speed (1500 mm/s) as the tracks do not overlap. The overlap between track 1 and track 2 is maximal at slice 3, where the laser turns from track 1 to track 2. Similarly, the overlap between track 2 and track 3 is maximal at slice 1. The difference in the overlap between different regions was very small in this case.

The melt pool boundaries obtained from three different regions, as shown in Figure 9a, were achieved for a 1 mm scan length and are shown in Figure 10. There was a noticeable difference between the melt pool boundary obtained with 0.5 mm and 1.0 mm scan lengths. The melt pool boundary obtained from slice 1 shows the overlap between track 2 and track 3, where the laser turns. In the laser-turn region, the melt pool of the second track may not solidify completely when the laser scans the third track. Hence, the melt pool from the third track will merge with the melt pool from the second track. However, there was no such overlap between track 1 and track 2, as the region includes the beginning of the first track and it solidifies when the melt pool from the second track reaches the beginning of the first track. The melt pool profile obtained in the middle of the track (slice 2) showed a more consistent overlap between the tracks. This is different from the overlaps observed in slice 1, and the merging of the melt pool from two tracks was not evident in this region. The merging of the melt pools depends on how long the melt pool lasts before complete solidification. The melt profile is inclined towards the previous melt pool, which may be due to the residual heat from the previous track. The melt pool profile obtained for slice

3 shows a higher overlap between track 1 and track 2. Again, the higher overlap at the laser turn is due to the residual heat and unsolidified melt pool. The laser turns from laser 1 to laser 2 in this region, which resulted in a larger overlap. Additionally, the melt pool obtained when using 1500 mm/s shows three distinct tracks, with no or minor overlap between the tracks. This is due to the use of higher hatch spacing in relation to the track width resulting from using 1500 mm/s at 195 W laser power.

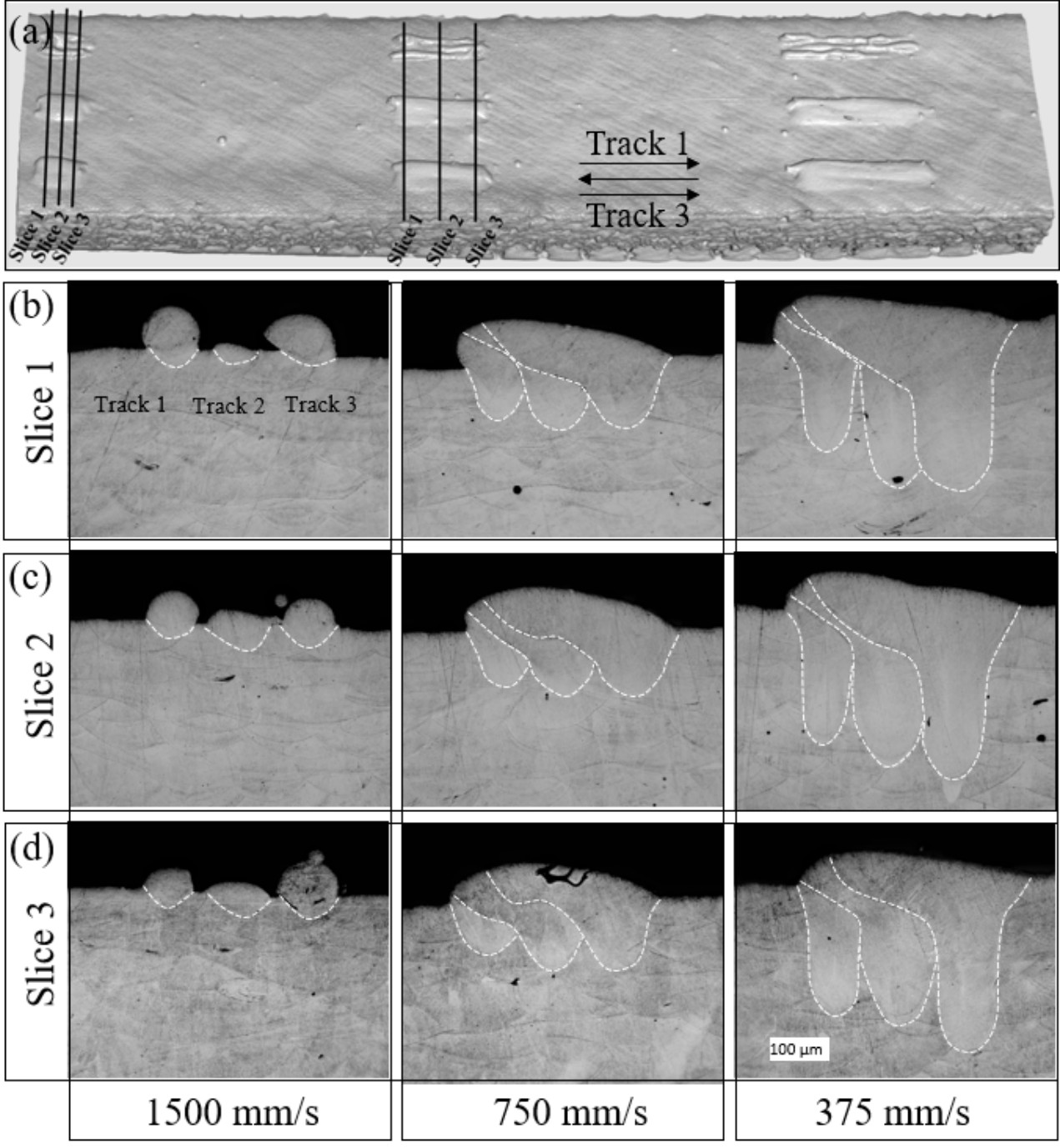

**Figure 9.** (**a**) Location of the three sections, and transverse melt pool boundary from (**b**) slice 1, (**c**) slice 2, and (**d**) slice 3 of 0.5 mm scan length formed with 195 W laser power, 120 μm hatch spacing and scanning speeds of 1500 mm/s (left), 750 mm/s (middle), and 375 mm/s (right).

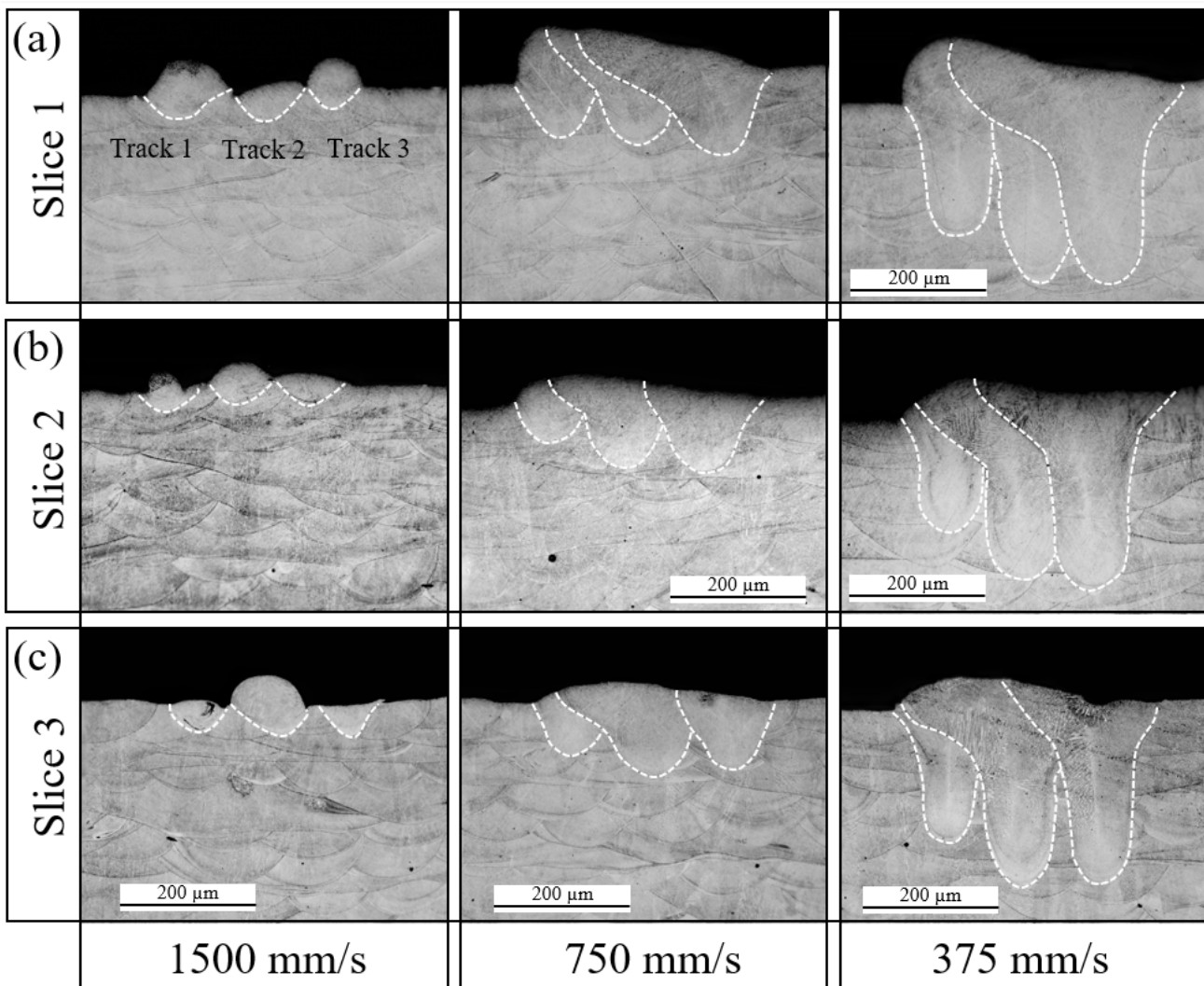

**Figure 10.** Transverse melt pool boundary of 1.0 mm scan length from (**a**) slice 1, (**b**) slice 2, and (**c**) slice 3 formed with 195 W laser power, 120 μm hatch spacing and scanning speeds of 1500 mm/s (left), 750 mm/s (center), and 375 mm/s (right).

The transverse melt pool boundary was also obtained from the two-layer sample. The multi-track experiment revealed the existence of a huge variation in the surface height along the longitudinal and transverse directions. This will cause the variation in the effective powder layer thickness for the second layer scanning, which in turn affects the laser penetration and the melt pool formation. Hence, metallography was performed to detect the differences in the melt profile between the first layer and the second layer. Figure 11 shows the transverse melt pool boundary obtained from the 1 mm sample. The images correspond to the slice 1 location discussed in the multi-track metallography. The melt pool boundaries formed with the two layers are distinct, which shows the similarity in the melt pool between the two layers due to the use of the same scanning strategy. However, the surface profile of the first layer affected the melt pool depth, which is significant in the conduction mode melting (Figure 11b). Lack-of-fusion and keyhole pores were also observed for the incomplete melting parameter (Figure 11a) and the keyhole parameter (Figure 11c), respectively. The lack-of-fusion pores formed between the track, while the keyhole pore formed inside the track.

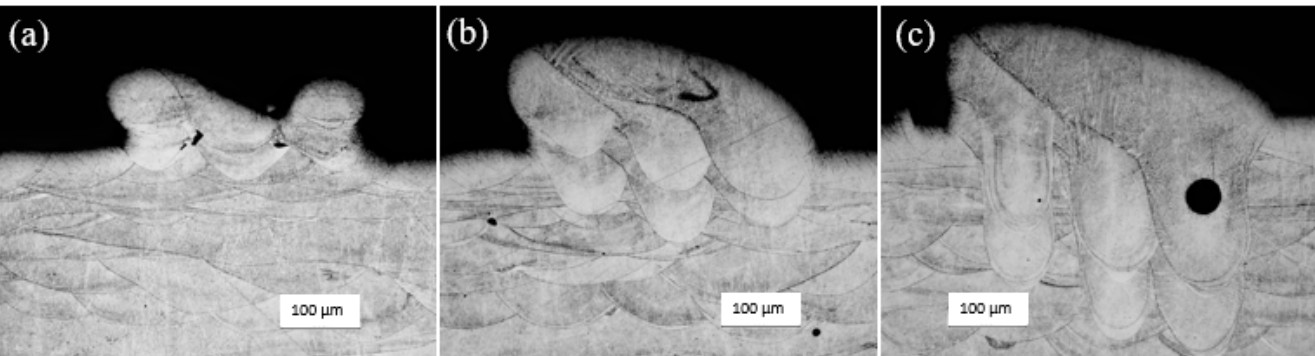

**Figure 11.** Micrographs of two-layers formed with 195 W laser power, 120 µm hatch spacing, (**a**) 1500 mm/s (**b**) 750 mm/s, and (**c**) 375 mm/s.

### 4.2. Numerical Results

The experimental results showed that the residual heat has a considerable effect on the melt pool boundary and the surface formation. The residual heat may depend on the properties of the material, which affect the life of the melt pool. Hence, numerical simulations were performed to understand the effect of residual heat on the development of the melt pool. In addition, the variation in the actual powder layer thickness due to the variation in the surface height of the first layer was investigated.

### 4.2.1. Multi-Track Results

Figure 12 presents the temperature contours of the multi-track simulations performed with 0.5 mm and 1 mm scan lengths. The laser power, scan speed and hatch spacing used in the simulation were 195 W, 750 mm/s, and 120 µm, respectively. The length of the melt pool was around 1 mm with the used laser parameters. Hence, the effect of the residual heat was significant in the case of 0.5 mm. As the laser began to scan the third track, the melt pool from the previous two tracks did not solidify in the case of 0.5 mm. However, the melt pool formed at the first track solidified when the laser started to scan the third track in the case of the 1 mm scan length. The lifetime of the melt pool highly affects the microstructure formed during the L-PBF process.

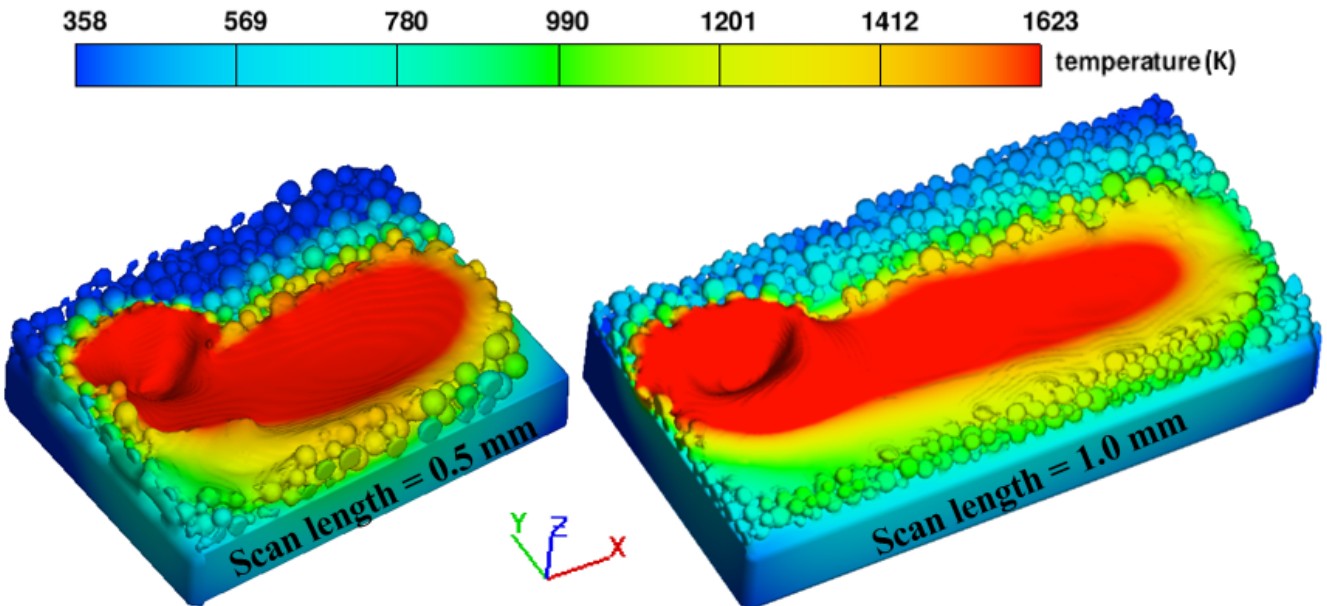

**Figure 12.** Temperature contours of the multi-track formed with 195 W laser power, 750 mm/s scan speed, 120 µm hatch spacing, and different scan lengths.

Figure 13 presents the transverse melt pool at different locations. The three slices discussed above (Figure 9) were also analyzed in the simulation. The melt pool obtained from the first track was indifferent at different locations, and the melt pool was mostly symmetrical. At the first location (slice 1), the melt pool formed with the second track was not affected by the first track as the temperature distribution shows that the melt pool had already solidified. However, the laser turns from track 2 to track 3 near location 1. Hence, the melt pool formed in track 3 scanning merged with the unsolidified melt pool from track 2. Similarly, the change in the shape of the melt pool between the successive tracks at the center of the tracks was obtained. As the laser reaches the center of the second and third tracks, the melt pool at the center of the previous tracks has already solidified. Hence, the melt pools do not overlap as they do in the laser-turn regions. Moreover, the shape of the melt pool is similar for the second and third tracks due to the similar effect of residual heat. Hence, it may be said that the melt pool was steadier along the transverse direction at the center of the 1 mm long back-and-forth scanning. The melt pool profile obtained at location 3 with laser turnaround from track 1 to track 2 is also presented. Again, the overlap between the tracks shows similar behavior to that of the melt pool profile obtained at location 1. As the laser scans the second track, the melt pool that formed merges with the unsolidified melt pool from the first track. However, the third track was not affected by the residual heat.

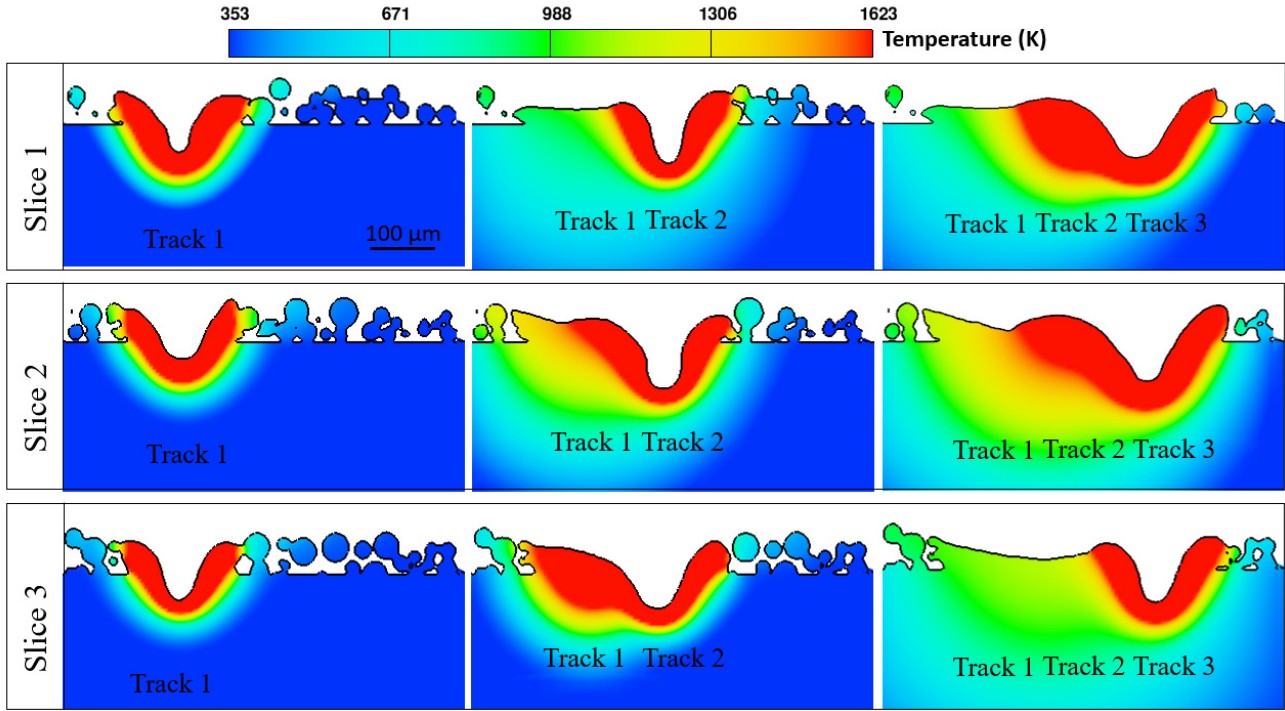

**Figure 13.** Temperature contours to observe the melt pool at different locations.

Figure 14 shows the top view of the three tracks and the melt profile obtained from three locations. The transverse melt region from slice 1 and slice 3 show the difference in the melt pool boundary in the laser-turning area, which can be observed from the temperature profile shown in Figure 13. The melt pool overlap area is higher when there is laser turning as seen in the experiment. The melt pool at the center of the tracks shows the gradual increase in the melt pool depth from track 1 to track 3.

The analysis of the transverse melt pool revealed the effect of residual heat and the unsolidified melt pool on the melt pool boundary. Figure 15 presents the longitudinal view of the simulation performed with 195 W, 750 mm/s, and 80 μm hatch spacing. The effect of the scan length on the development of the melt pool and the melt pool depression was analyzed. There was a significant increase in the depression at the beginning of the second

track due to the residual heat. Although the constant energy input was maintained, the residual heat from the previous track led to the formation of a deeper melt pool at the beginning of the second track. As the laser traveled to the middle of the track, the effect of the residual heat lessened. However, the effect of residual heat was still present, and the depth of the melt pool increased. Again, the depth of the melt pool increased significantly as the laser changed direction for track 3 scanning. Besides, the transverse melt regions taken at different regions from the 0.5 mm and 1.0 mm cases, showed that the effect of the residual heat in the laser-turn region was similar. The major difference between the two cases was the melt pool boundary at the center of the scan track, which was 0.25 mm for 0.5 mm scan length and 0.5 mm for 1.0 mm scan length. The depth of the melt pool increased significantly with an increasing number of tracks for 0.5 mm, due to the residual heat. In contrast, the melt pool depth increased by the small amount of 1.0 mm. In this regard, the residual heat had a significant effect on the smaller tracks.

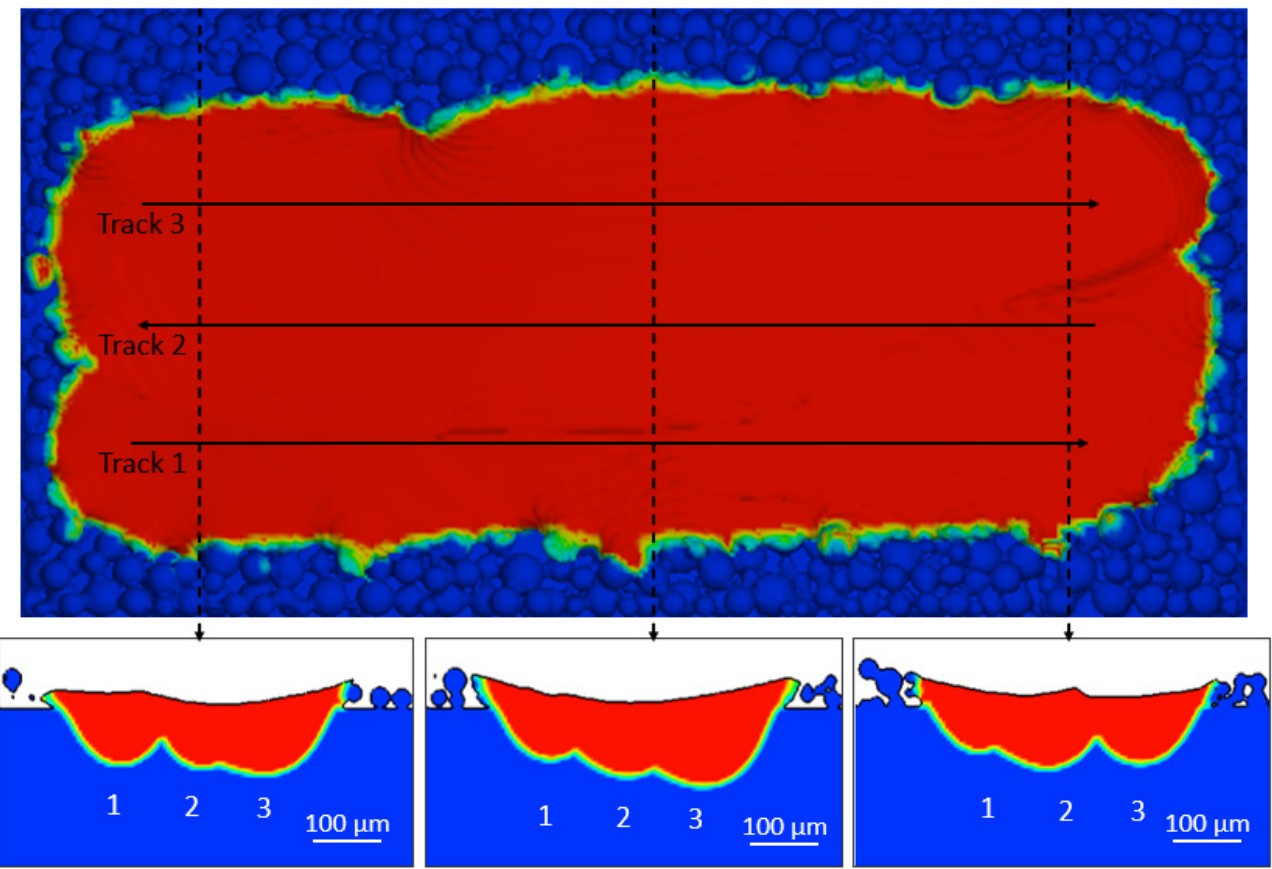

**Figure 14.** Melt profile formed at different regions along the scan direction.

### 4.2.2. Multi-Layer Results

The multi-track simulation performed over the solid substrate showed the limitation of the mesh-based numerical model. The powder dynamics and its effect on the single-track morphology were not captured. However, the variation in the track surface due to the shrinkage was evident. The height of the track decreased from the start of the scan to the end, and from the first track to the third track. Consequently, Figure 16 shows the variation in the actual powder layer thickness obtained from the second layer DEM simulation. The green region is the mesh obtained from the first layer simulation, and the blue region shows the powder particles being deposited over the first layer surface during the second layer DEM simulation. The resulting differences in the powder distribution will affect the subsequent melt pool formation.

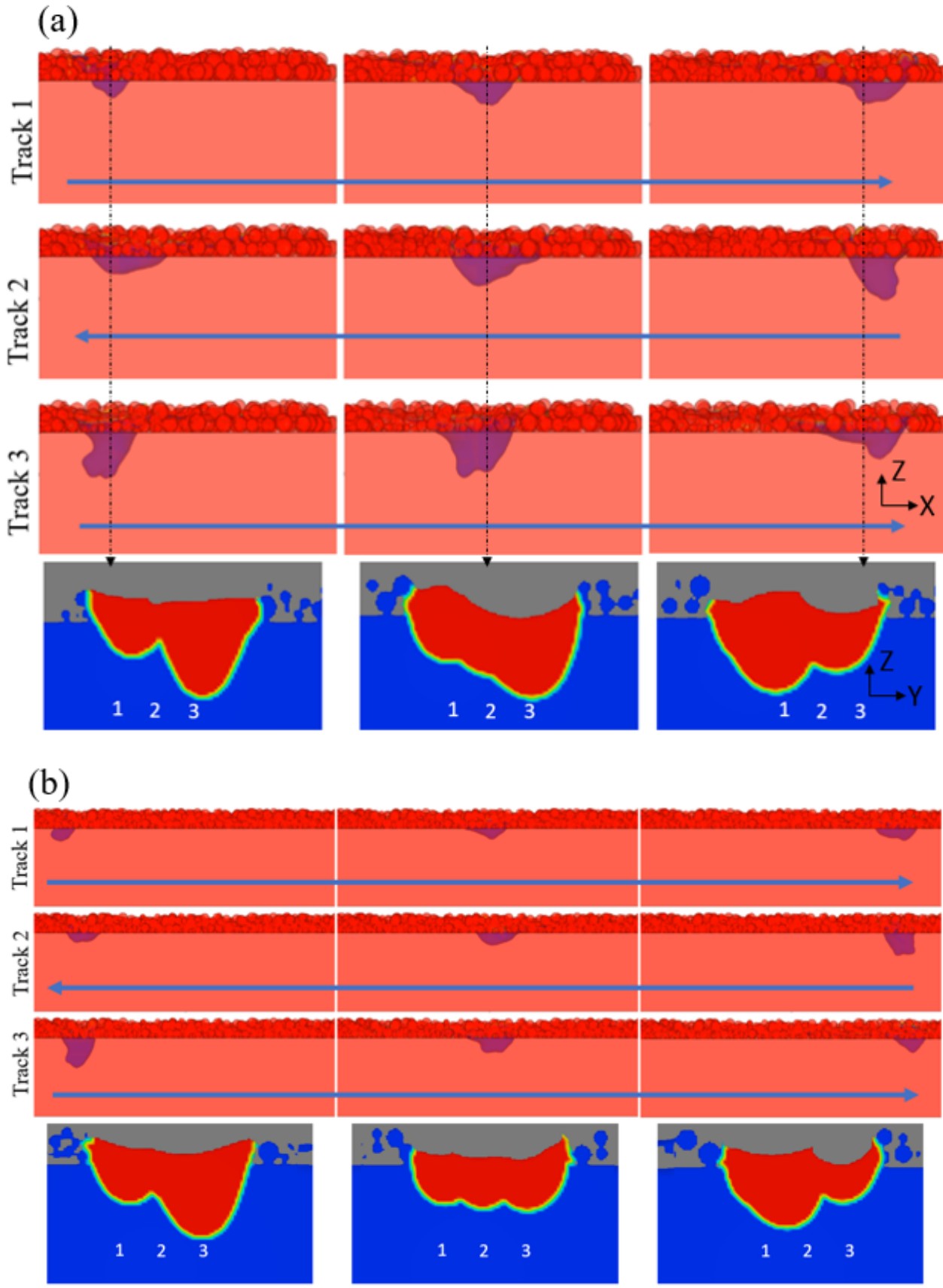

**Figure 15.** Evolution of melt pool and vapor depression with 195 W laser power, 750 mm/s scan speed, 80 μm hatch spacing, and scan lengths of (**a**) 0.5 mm and (**b**) 1.0 mm.

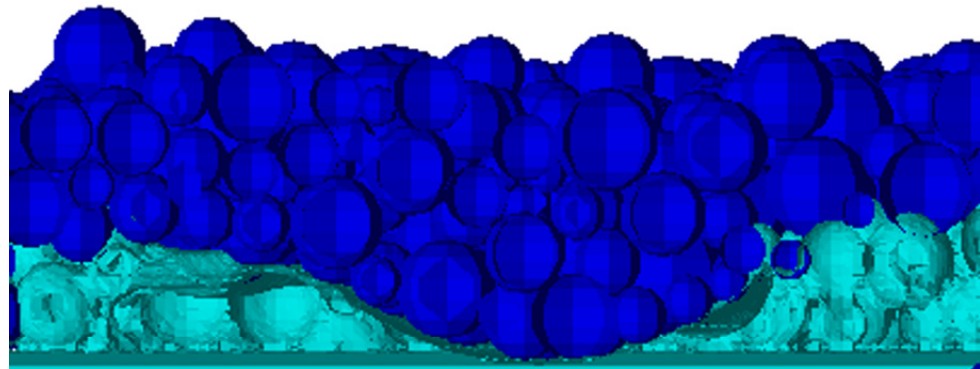

**Figure 16.** Variation in the actual powder layer thickness due to the variation in the surface height of the substrate layer.

Figure 17 presents the transverse melt profile obtained at three different locations from the two-layer simulation. The melt pool boundary of the first layer, the powder distribution after the second layer DEM simulation, and the second layer melt pool boundary were compared at different locations. The actual powder layer thickness varied across the second layer, as seen from the three slices taken at different locations. The powder distribution affected the second layer melt pool boundary. The increase in the effective layer thickness affected the melt pool beneath the previously deposited layer. However, the trend of the melt pool boundary was similar to the first layer. There was a huge overlap between the tracks in the laser-turn region. Besides, the second layer prediction depends on the first layer surface morphology and the resultant powder distribution. As the simulation model predicted a surface profile that was contrary to the experimental result, the resultant melt pool boundary was not comparable to the experimental melt pool boundary.

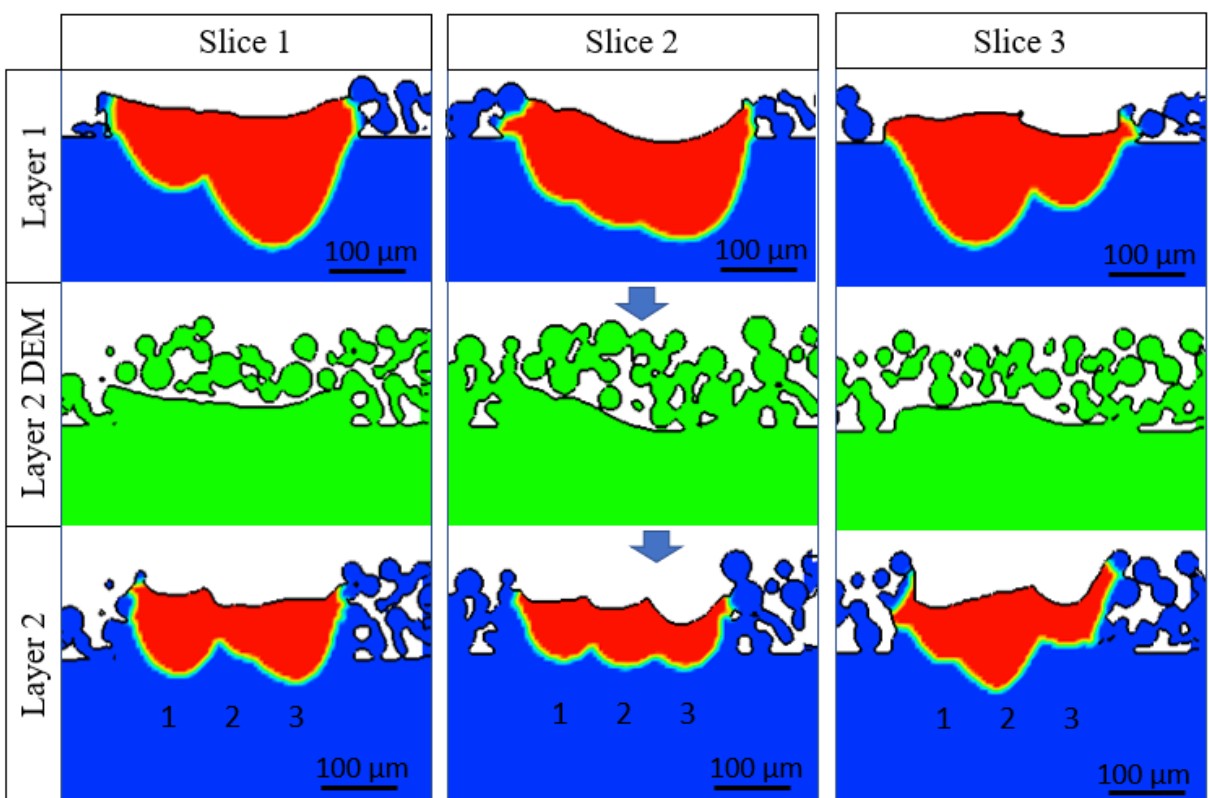

**Figure 17.** Second layer melt pool boundary obtained with 195 W laser power, 750 mm/s scan speed, 80 μm hatch spacing and 0.5 mm scan length.

The second layer simulation was also performed for low energy density cases. Again, the transverse melt profile from three different regions is presented in Figure 18. The use of lower energy density shows the formation of the lack-of-fusion pores. The formation of the lack-of-fusion pores is apparent in the first layer simulation. In addition, the surface morphology worsened in the second layer. Such behavior was also observed in the experiment, as the surface became very distorted in the second layer scanning.

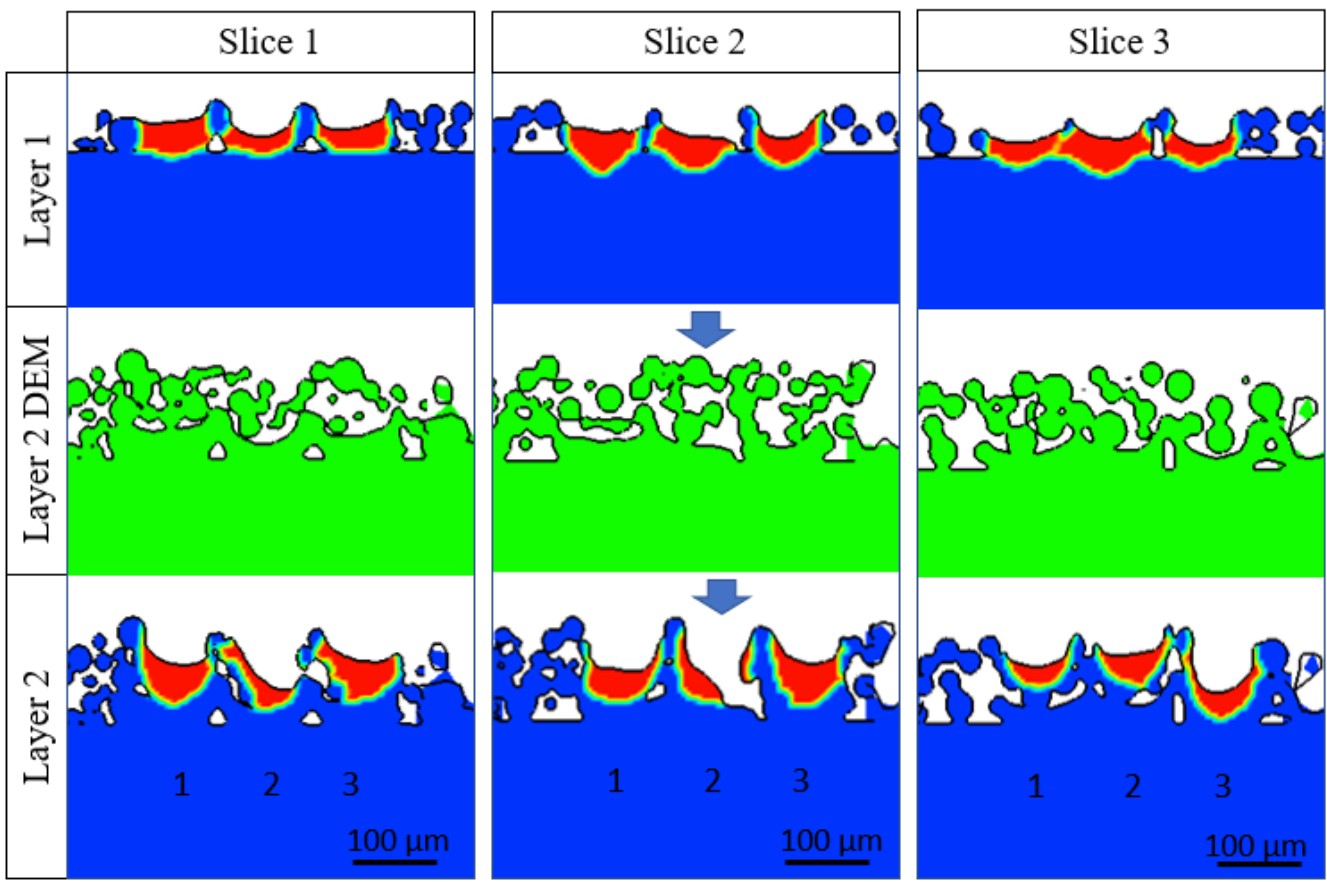

**Figure 18.** Second layer melt pool boundary obtained with 195 W laser power, 1500 mm/s scan speed, 120 μm hatch spacing and 0.5 mm scan length.

## 5. Conclusions

In this paper, a combined experimental and numerical study was performed to understand the effect of scan length on the residual heat and melt pool behavior during multi-track and multi-layer fabrication. EOS M270 was used to fabricate IN625 multi-track samples with scan lengths of 0.5 mm, 1 mm, and 1.5 mm. A white light interferometer was initially used to measure the surface profile. Thereafter, metallography was performed to reveal the transverse melt profile and observe the melt pool shape variations along the laser travel direction. In addition, a thermo-fluid numerical model was developed to understand the physics behind the discrepancies in the melt pool in different regions of the sample. The analyses of the results led to the following conclusions.

- The transverse melt pool boundary obtained from the laser-turn region showed a significant overlap between the tracks, both in the experiment and simulation. The numerical analysis showed that the overlap was due to the merging of the melt pools from the two successive tracks. Besides, the overlap region was significantly higher for a 0.5 mm scan length compared to 1 mm.

- The hatch spacing also had a significant effect on the formation of the subsequent melt pool boundary due to the presence of residual heat. Apart from the increase in the overlap, the depth of the melt pool increased significantly with lower hatch spacing.
- The surface profile of the first layer affected the actual powder layer thickness in the second layer, which in turn affected the second track formation in the second layer. The micrograph showed that the gap between the two-layer boundaries was higher for the region with a higher first-layer surface height.

## 6. Future Work

The scan length has a significant effect on the surface morphology and the microstructure. Hence, parameter customization based on the scan area is necessary. A parameter that is suitable for large-part fabrication may not be suitable for thin feature printing. Future work on this topic includes parameter variation according to the scan region to reduce the variability in the quality of the part.

**Author Contributions:** Conceptualization, S.S. and K.C.; methodology, S.S. formal analysis, S.S.; investigation, S.S.; resources, K.C.; writing—original draft preparation, S.S.; writing—review and editing, K.C.; supervision, K.C.; project administration, K.C.; funding acquisition, K.C. All authors have read and agreed to the published version of the manuscript.

**Funding:** This research is partially supported by NSF, Manufacturing Machines & Equipment Program (Award # 1662662).

**Data Availability Statement:** The raw/processed data required to reproduce these findings cannot be shared at this time due to technical or time limitations.

**Conflicts of Interest:** The authors declare no conflict of interest.

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
