# Peer review of "Residual Heat Effect on the Melt Pool Geometry during the Laser Powder Bed Fusion Process"

_jmmp, doi:10.3390/jmmp6060153_

Round 1
Reviewer 1 Report
1. In the introduction, the existing problems should be summarized after the analysis of the current research status, which leads to the necessity of this paper.
2. Hatch spacing shall be defined and explained when it appears for the first time (it is better to use the figure).
3. Line 128 ”literature[]”.
4. The accuracy of numerical approach needs to be verified.
5. How to distinguish the boundaries of the three tracks in Figure 10.11, especially “750mm/s” and “375mm/”.
6. The progressive influence relationship in Figure 18 is explained in detail, the specific influence of powder distribution on the second layer melt pool boundary is difficult to see.
Reviewer 2 Report
Overall review
1.The overall logic of the article is not strong, but the necessary description of the experiment.
2.The effect of residual heat on the geometry of the melt pool does not indicate what effect it will have in practical use.
3.The references in the introduction are lack of logical.
4.The experimental steps were not explained separately, and the description was unclear.
5. Pictures and tables did not express the experiment well.
6. The article uses too much literature content, which seems to be an experiment based on the literature content, without a good expression of what the author actually did.
7. The layout of pictures and tables in the text is unreasonable.
Questions
1. What aspects will the geometry of the melt pool affect in practical applications?
2. In the text in lines 217 and 218, when scanning at a higher speed (1500mm/s), the tracks will not overlap because the track width does not match the hatch spacing. In lines 243-245, there is little or no overlap of the tracks because of the use of higher hatch spacing. There is no overlap between the tracks in the two statements. Are the two statements contradictory?
3. In lines 237 and 238, what is the lifetime of the melt pool?
Reviewer 3 Report
In this paper, the authors studied the effect of residual heat on the melt pool geometry during the LPBF process through both experiments and simulations. After reading the whole content, I regret to say that I don’t see enough novelty and impact in this paper. The phenomena reported here are well-known and well-expected. The writing also consists of many logical errors, typos, and grammatical errors. The Introduction failed to clearly point out the knowledge gaps in the field. The Results should refer to specific figures as much as applicable. Overall, I do not think the paper adds enough intellectual merit to the community. The writing style also needs to be reviewed and edited by professionals with technical writing backgrounds.
Round 2
Reviewer 1 Report
-
The manuscript has been modified according to the suggestions, and the quality of the manuscript has been improved.
Author Response
We thank the reviewer for the valuable comments. The authors have made further grammatical corrections.
Reviewer 2 Report
I think the paper can be accepted for publication now.
Author Response
We thank the reviewer for the valuable comments. The paper did consist of few grammatical errors which have been corrected in the new version.
Reviewer 3 Report
I regret to say that I still believe this paper adds little intellectual merit to the community. The findings and conclusions are well-known and well-expected.
Author Response
We thank the reviewer for the comments. The revised paper consisted of many grammatical errors. In the newer version, these grammatical errors have been corrected.